

# Multi-model comparison of urban heat island modelling approaches

Jan Karlický[1], Peter Huszár[1], Tomáš Halenka[1], Michal Belda[1], Michal Žák[1], Petr Pišoft[1], and Jiří Mikšovský[1]

[1]Department of Atmospheric Physics, Faculty of Mathematics and Physics, Charles University, Prague, V Holešovičkách 2, 180 00 Prague 8, Czech Republic

*Correspondence to:* J. Karlický (Jan.Karlicky@mff.cuni.cz)

**Abstract.**

Cities are characterized by different physical properties of surface compared to their rural counterparts, resulting in specific regime of the meteorological phenomenon. Our study aims to evaluate the impact of typical urban surfaces on the central-European urban climate in several model simulations, performed with the WRF and RegCM models. The specific processes occurring in the typical urban environment are described in the models by various types of urban parametrizations, greatly differing in complexity. Our results show that all models and urban parametrizations are able to reproduce the most typical urban effect, the summer evening and nocturnal Urban Heat Island, with the average magnitude of 2–3 °C. The impact of cities on the wind is clearly dependent on the urban parametrization employed, with more simple ones unable to fully capture the wind speed reduction induced by the city. In the summer, a significant difference in the boundary layer height (about 25 %) between models is detected. The urban induced changes of temperature and wind speed are propagated into higher altitudes up to 2 km, with a decreasing tendency of their magnitudes. With the exception of the summer daytime, the urban environment improves the weather conditions a little with regard to the pollutant dispersion, which could lead to the partly decreased concentration of the primary pollutants.

## 1 Introduction

From a global point of view, cities represent small and separated areas with significantly different surface properties compared to the surrounding rural ones. Considering the fact that more than half of the human population live in cities and the number is still increasing (United Nations, Department of Economic and Social Affairs, 2014), the investigation of impact of urbanized areas on environment, especially on atmospheric conditions, becomes of crucial importance (Folberth et al., 2015). There are many ways by which urban areas influence the atmosphere: from directly impacting the air composition and consequently the radiative balance and climate (Huszar et al., 2016; Huszár et al., 2016) to the meteorological forcing cities represent, best known in terms of the urban heat island effect. The concept of Urban Heat Island (UHI) was introduced many decades ago (Oke and Maxwell, 1975) and embodies that, due to different thermal, radiative, hydrological and mechanical properties of urban surfaces and due to anthropogenic thermal resources, temperatures in the city centres are a few degrees higher than in the city surroundings. This is true for averaged temperature difference, but the impact is not uniformly distributed across the day and at specific times of day (typically evening and night), the difference can exceed 10 °C (Oke, 1982).



Other observation and model based studies (e.g. Theeuwes et al., 2015; Lee et al., 2011; Huszar et al., 2014) describe that not only temperature is affected by urban surfaces, but the wind speed is also significantly altered (Roth, 2000; Klein et al., 2001; Hou et al., 2013). Urban surfaces further influence the structure of the boundary layer as well (Angevine et al., 2003), along with the height of the planetary boundary layer (PBLH), which is very important from an air-quality perspective. It was also shown by Huszar et al. (2014) that whole regions can be affected by urban meteorological effects and the magnitude of the temperature increase can be compared to the magnitude caused by the climate change. It was also of interest to investigate the possible modifications of UHI in a changing climate. In the future, due to the fact that global temperature is predicted to rise, together with cities growing and cities population rising, their inhabitants could be affected by more intensive, frequent and longer heat waves, as described by Meehl and Tebaldi (2004). To prevent these dangerous scenarios, many mitigation measures are proposed and tested using climate models (e.g. Fallmann et al., 2016), but the climate models are still, despite their steady advancement, burdened by many uncertainties and inaccuracies, so further development, evaluation and inter-model comparisons are still needed.

Last decade, many validation studies appeared, where numerical weather prediction and/or regional climate models are coupled to various types of Urban Canopy Models (UCM, e.g. Chen et al., 2011; Lee et al., 2011; Liao et al., 2014) in order to capture different urban processes on local and regional scale with an effort to describe especially the urban heat island more accurately. These studies applied one of the wide range of approaches to parametrize the urban meteorological phenomenon, from simply modifying the surface parameters to represent an average value corresponding to urban surface – so-called Bulk parametrization (described e.g. in Chen et al., 2011), over models considering a single urban layer and an idealized street canyon (e.g. Single-Layer Urban Canopy Model – SLUCM; Kusaka et al., 2001), to more sophisticated multi-layer models of urban environment with ability to include different heights of buildings in the city and to resolve the vertical structure of the urban canyon (e.g. Building Environment Parametrization – BEP; Martilli et al., 2002).

Due to the fact that not only the temperature is affected by different urban surface, but also other variables, including ones important from the air-quality perspective (e.g. boundary layer structure, turbulence), it is necessary also to study the impact of the urban induced meteorological changes on air-quality. Studies, evaluating this impact, usually use coupled meteorological, urban canopy and chemical-transport models (Liao et al., 2014; Fallmann et al., 2016; Huszár et al., 2018) and they conclude that inclusion of the urban meteorological effects has important impact to final species concentrations, both primary and secondary ones.

Despite of the urban canopy parametrization progress, and how often they are implemented in the surface schemes, there are still large uncertainties in modelling the mesoscale meteorological conditions within the urban environment. These primarily emerge from the uncertainties given by the choice of the urban canopy scheme. Moreover, additional sources of uncertainties include: the driving surface model and the meteorological model itself, uncertainties in the driving meteorological data and the uncertainty of the choice of the urban parameters. Urban models differ in the processes considered and in the way how they are implemented. It is widely accepted that complex urban parametrizations that include more processes are able to capture the urban phenomenon more accurately. However, recently Best and Grimmond (2015) showed that this is not a general rule. They concluded that there is a 'need to balance the requirement for complexity within models against what is actually required for a



model to be fit for purpose' and even a simple approach suffices if the most relevant processes are included. This implies that the general goal should not only be the continuous development of more complex urban modelling approaches, but also their inter-comparison and contrasting with simpler techniques. Such inter-comparisons are important in terms of driving models and meteorological conditions as well. Finally, as background climate is a strong factor influencing the UHI and other urban

effects (Zhao et al., 2014), long-term simulations are necessary to reveal the long-term variability of these effects.

Our study aims to contribute to the above mentioned works by looking at the uncertainty of UHI modelling associated with selection of the driving meteorological model and the underlying UCM. Another goal is to investigate the benefits of more sophisticated urban parametrizations with respect to simple approaches in the light of the conclusions of Best and Grimmond (2015). The work will focus not only on averaged values but on variability and extreme values as well that are crucial in

assessing the urban impact on environment and human life. The work also presents the urban impact on poorly observed or immeasurable variables in the city and its surroundings, e.g. temperature and wind profiles or boundary layer height. Lastly, it evaluates also air-quality related meteorological quantities that directly influence pollutant dispersion, which is an important precondition for urban air quality. All these issues will be evaluated in long-term perspective using 10 yr long simulations.

## 2 Methods

### 2.1 Models, Urban parametrizations

In our study, two meteorological models are used, the Weather Research and Forecasting model (WRF) and the regional climate model version 4 (RegCM4). The WRF model (Skamarock et al., 2008) is a mesoscale, non-hydrostatic limited area meteorological model, developed originally for weather prediction, but also widely used as a regional climate model. All simulations used in this study are performed by WRF version 3.7.1, which enables connection with Bulk, SLUCM, BEP and

BEP/BEM urban parametrizations (description below).

The second model used in this study, RegCM4, is described in detail by (Giorgi et al., 2012). RegCM4 is a mesoscale, non-hydrostatic (hydrostatic dynamics included) limited area regional climate model developed at the International Centre for Theoretical Physics (ICTP). Simulations performed for this study were executed by RegCM model version 4.4, which offers connection with two different models of land-surface processes: Biosphere-Atmosphere Transfer Scheme (BATS; Dickinson

et al., 1993) and the more detailed scheme called Community Land Model version 4.5 (CLM4.5; Lawrence et al., 2011; Oleson et al., 2013). Both land-surface schemes include parametrizations of the urban canopy detailed further.

The simplest way to capture the impact of urban surfaces is to adapt values of parameters representing surface characteristics such as roughness length, surface albedo, heat capacity, soil thermal conductivity and green-vegetation fraction for urban environment conditions, which represent a zero-order effect of urban surfaces (Chen et al., 2011). By default, The WRF model

uses values of these parameters obtained by (Liu et al., 2006). This approach is most often called the Bulk parametrization (Chen et al., 2011; Lee et al., 2011), sometimes also referred to as a slab model (Kusaka et al., 2001; Kusaka and Kimura, 2004). In further, we will refer to it as the Bulk parametrization.

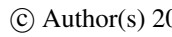



The Single-Layer Urban Canopy Model (SLUCM; Kusaka et al., 2001; Kusaka and Kimura, 2004) represents a next level of model approximation of the urban meteorological phenomena. Here, the city is assumed as an infinitely long street canyon with different prescribed orientations. This enables to include shadowing, reflections and trapping of radiation. Further, this model distinguishes temperatures of roofs, walls and roads and profiles within these surfaces. For wind, an exponential profile

is assumed. The total sensible heat flux from all surfaces, the total momentum flux and friction velocities are then returned back to the land-surface and boundary layer schemes of the driving models. The CLMU urban parametrization (Oleson et al., 2008), implemented in the RegCM model under the CLM4.5 land surface scheme, is also based on canyon representation of urban areas and is conceptually very similar to SLUCM.

The most sophisticated urban parametrization used in this study is the multi-layer urban canopy model called Building

Environment Parametrization (BEP; Martilli et al., 2002). Its main benefits lie in allowing the computation of vertical profiles of temperature, momentum and turbulent kinetic energy within urban canyon explicitly, taking into account the vertical distribution of the sources and sinks of heat, momentum and moisture, and its direct interactions with the boundary layer parametrization.

The BEP urban model can be improved by using a scheme that computes the energy exchange between the atmosphere

and the interior of the building. The WRF model implements such scheme, called Building Energy Model (BEM) developed by Salamanca et al. (2009). In opposite to SLUCM, which enables only simple inclusion of the Anthropogenic Heat (AH) with constant daily profile and annual profiles, the BEM allows both double-sides energy exchange and computing of the AH depending on specific weather conditions, which makes the AH flux more realistic. To achieve this, the BEM takes into consideration the following processes: heat and radiation transfer between indoor and outdoor area, heat and radiation transfer

between indoor walls and floors, indoor heat generation by human bodies and machines and finally ventilation, heating and air conditioning (Chen et al., 2011).

## 2.2 Experimental setups

All simulations performed within this study were run on a $160\times120$ domain centred over Central Europe, with $10\,\mathrm{km}\times10\,\mathrm{km}$ horizontal resolution (Fig. 1). In vertical direction, 30 and 23 levels are considered for WRF and RegCM, respectively. The

model top is 50 hPa in both cases. The simulation time-span is 10 years (2001–2010), without any restart or nesting procedure. As meteorological boundary conditions, the ERA-interim (Dee et al., 2011) dataset is used. For static geographic data, standard WRF and RegCM input USGS-based data are used. The standard WRF land-use input includes only one urban category type. In case of RegCM experiments with the CLM4.5, urban land-use percentage was derived from the 0.05° resolution LandScan2004 data described by Jackson et al. (2010). They provide urban canyon parameters and surface characteristics. For the RegCM with

the BATS/SLUCM, 2 km × 2 km sub-grid was applied for surface processes. Each sub-gridbox is considered to be covered by one land-use type. Urban and sub-urban categories are considered. In order to have consistent land-use data for both RegCM setups, we set the number of urban and sub-urban sub-gridboxes to correspond to the fraction defined in case of the CLM4.5 urban setup. Urban canopy parameters used by urban schemes are adapted to urban environment in Prague and are listed in Table 1.



| Parameter | Unit | Value | Used in |
|---|---|---|---|
| Building height | m | 17.5 (20.0, 15.0) | SLUCM |
| Roof width | m | 17.5 (20.0, 15.0) | SLUCM, BEP+BEM |
| Road width | m | 17.5 (15.0, 20.0) | SLUCM, BEP+BEM |
| Anthropogenic heat | $W\,m^{-2}$ | 35.5 (50.0, 20.0) | SLUCM |
| Urban fraction | – | 0.8 (0.9, 0.7) | SLUCM, BEP+BEM |
| Heat capacity of roof | $J\,m^{-3}K^{-1}$ | $0.776 \cdot 10^6$ | SLUCM, BEP+BEM |
| Heat capacity of building wall | $J\,m^{-3}K^{-1}$ | $1.02 \cdot 10^6$ | SLUCM, BEP+BEM |
| Heat capacity of ground (road) | $J\,m^{-3}K^{-1}$ | $1.7 \cdot 10^6$ | SLUCM, BEP+BEM |
| Thermal conductivity of roof | $J\,m^{-1}s^{-1}K^{-1}$ | 0.8 | SLUCM, BEP+BEM |
| Thermal conductivity of building wall | $J\,m^{-1}s^{-1}K^{-1}$ | 1.28 | SLUCM, BEP+BEM |
| Thermal conductivity of ground (road) | $J\,m^{-1}s^{-1}K^{-1}$ | 0.6 | SLUCM, BEP+BEM |
| Surface albedo of roof | – | 0.30 | SLUCM, BEP+BEM |
| Surface albedo of building wall | – | 0.27 | SLUCM, BEP+BEM |
| Surface albedo of ground (road) | – | 0.16 | SLUCM, BEP+BEM |
| Surface emissivity of roof | – | 0.70 | SLUCM, BEP+BEM |
| Surface emissivity of building wall | – | 0.88 | SLUCM, BEP+BEM |
| Surface emissivity of ground (road) | – | 0.92 | SLUCM, BEP+BEM |
| Thickness of each roof layer | m | 0.05, 0.05, 0.05, 0.05 | SLUCM |
| Thickness of each building wall layer | m | 0.05, 0.05, 0.05, 0.05 | SLUCM |
| Thickness of each ground (road) layer | m | 0.05, 0.25, 0.50, 0.75 | SLUCM |
| Roughness length for momentum over roof | m | 0.01 | BEP+BEM |
| Coefficient of performance of the A/C systems | – | 3.5 | BEP+BEM |
| Coverage area fraction of windows in the walls of the building | – | 0.2 | BEP+BEM |
| Thermal efficiency of heat exchanger | – | 0.75 | BEP+BEM |
| Target Temperature of the A/C systems | K | 298 (± 0.5) | BEP+BEM |
| Target humidity of the A/C systems | $Kg\,Kg^{-1}$ | 0.005 (± 0.005) | BEP+BEM |
| Peak number of occupants per unit floor area | $person\,m^{-2}$ | 0.01 | BEP+BEM |
| Peak heat generated by equipments | $W\,m^{-2}$ | 18.00 | BEP+BEM |
| Street direction | degrees from North | 0, 90 | BEP+BEM |
| Building heights | m | 10, 15, 20, 25 | BEP+BEM |
| Building heights percentage | % | 10, 40, 40, 10 | BEP+BEM |
| Diurnal AH profile | | 0.16 0.13 0.08 0.07 0.08 0.26 0.67 0.99 0.89 0.79 0.74 0.73 | |
| | | 0.75 0.76 0.82 0.90 1.00 0.95 0.68 0.61 0.53 0.35 0.21 0.18 | |
| Diurnal heating profile of heat generated by equipments | | 0.25 0.25 0.25 0.25 0.25 0.25 0.25 0.5 1.0 1.0 1.0 1.0 | |
| | | 1.0 1.0 1.0 1.0 1.0 1.0 0.5 0.25 0.25 0.25 0.25 0.25 | |

**Table 1.** Urban canopy parameters used in simulations. Values in brackets indicate different parameters used in RegCM simulations with the SLUCM scheme, in which urban and sub-urban categories are considered.



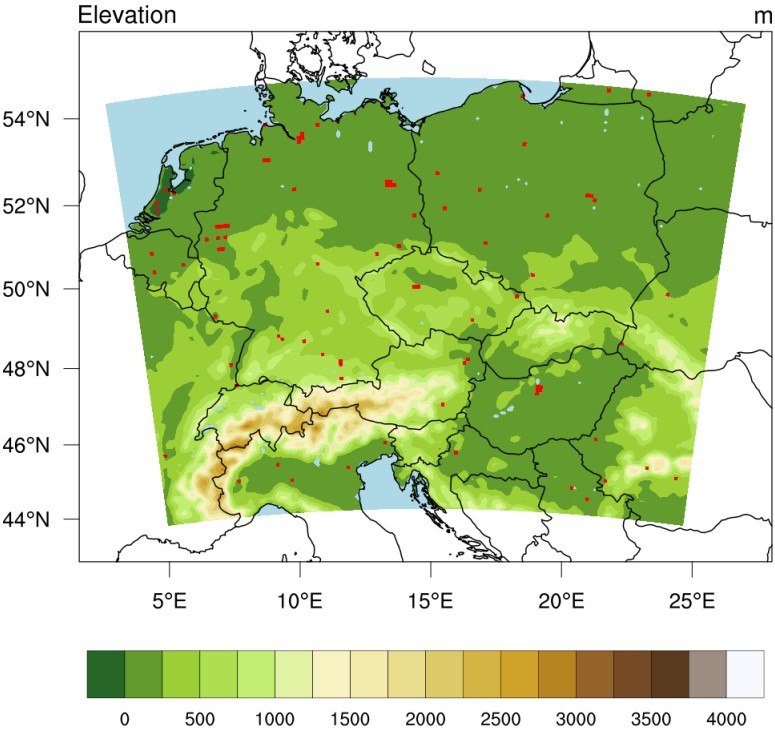

**Figure 1.** Position of the model domain with model topography in 10 km resolution (m). Grid-boxes with dominant urban land-use are marked by red color.

For the WRF model, simulations were performed with its hydrostatic version. For the radiation transfer, RRTMG scheme (Iacono et al., 2008) is used for both long- and short-wave spectrum. Microphysical processes are resolved by the Morrison double-moment scheme (Morrison et al., 2009). Further, the Noah Land Surface Model (Chen and Dudhia, 2001) is chosen for the description of the land-surface processes, surface layer processes are parametrized according to Janjić (1994), planetary

5 boundary layer is resolved by Mellor-Yamada-Janjic scheme (Janjić, 1994) and Tiedtke scheme (Tiedtke, 1989) for convection is used.

In case of the RegCM model, also the hydrostatic version is used. Radiation transfer is resolved by NCAR Community Climate Model Version 3 (CCM3; Kiehl et al., 1996). The Holtslag scheme (Holtslag et al., 1990) is applied to planetary boundary layer processes. For large-scale precipitation and convection, the SUBBEX (Pal et al., 2000) and the Grell schemes

10 (Grell, 1993) are used, respectively. As already noted, the RegCM model includes two different parametrizations of land-surface processes, specifically the more simple and older BATS (connected with SLUCM urban scheme), which is configured with 2 km×2 km sub-grid setting, and the more detailed CLM4.5 (connected with CLMU urban scheme).





| Experiment | Abbreviation | Model | Urban inclusion | Urban scheme | Land-surface model |
|---|---|---|---|---|---|
| WRF–woU | W–woU | WRF | No | – | Noah LSM |
| WRF–BULK | W–B | WRF | Yes | Bulk | Noah LSM |
| WRF–SLUCM | W–S | WRF | Yes | SLUCM | Noah LSM |
| WRF–BEP+BEM | W–BB | WRF | Yes | BEP+BEM | Noah LSM |
| RegCM–SLUCM–woU | R–S–woU | RegCM | No | – | BATS |
| RegCM–SLUCM | R–S | RegCM | Yes | SLUCM | BATS |
| RegCM–CLM4.5–woU | R–C–woU | RegCM | No | – | CLM4.5 |
| RegCM–CLM4.5 | R–C | RegCM | Yes | CLMU | CLM4.5 |

**Table 2.** Summary of simulation experiment setups.

The aim of this study is to determine the impact of cities on climate. To this end, two types of simulations are performed: firstly simulations with cities (or urban surfaces in general) using the unaltered geographic data and, secondly, simulations where urban surfaces are removed from geographic data and replaced by the dominating land-use category in its surroundings (crops in the majority of cases). Unfortunately, the models adopted do not allow for connection with all listed urban models. However, all allowed and meaningful combinations are considered. The summary of the simulations performed is provided by Table 2.

### 2.3 Observational data

For the basic validation of model outputs, the E-OBS (v. 12.0) dataset (Haylock et al., 2008) is used. E-OBS includes a $0.25° \times 0.25°$ resolution gridded data of temperature and precipitation for the whole modelled period. The ECAD data (Klein Tank et al., 2002) including station-based temperature means, maxima and minima are used for evaluating the temperature differences between city centres and their vicinity. Additionally, data supplied by the Czech Hydro-Meteorological Institute (CHMI) provide hourly temperatures as well as their daily averages and extremes for stations in both the centre and the vicinity of the Czech capital, Prague.

## 3 Results

### 3.1 Model validation

Fig. 2 shows the seasonal model biases for temperature means, maxima, minima and daily precipitation means (compared to the E-OBS data). In general, model biases depend mainly on the model type, eventually on the land-surface scheme; the choice of different urban parametrizations and the inclusion/exclusion of urban surfaces have only minor effect on the overall bias as only a small fraction of grid-points in the whole domain is covered by urban land-use type. The temperature means are reproduced by the WRF model with a bias up to around 1 °C. For the RegCM model coupled to BATS, biases are mostly



1–2 °C, but for the RegCM model combined with CLM4.5 land-surface model, they reach 2–4 °C. Temperature extremes are generally captured with even higher biases. In terms of average daily precipitation, the WRF model captures the annual cycle more accurately; the RegCM simulations exhibit large winter positive bias contrasting the E-OBS annual cycle. The values are also substantially overestimated in RegCM simulations, mainly in the ones with the CLM4.5 land-surface model.

Fig. 3 shows the spatial distribution of precipitation biases for all seasons and for WRF–SLUCM, RegCM-SLUCM and RegCM–CLM4.5 simulations. It is clearly visible that great winter overall biases in RegCM precipitation originate from mountainous regions (Alps). However, this overestimation does not occur in the summer season, leading to flipped annual cycle of domain-averaged precipitation in RegCM simulations, compared to the E-OBS precipitation cycle. On the other hand, WRF simulations tend to overestimate mountain area precipitation particularly in the summer season.

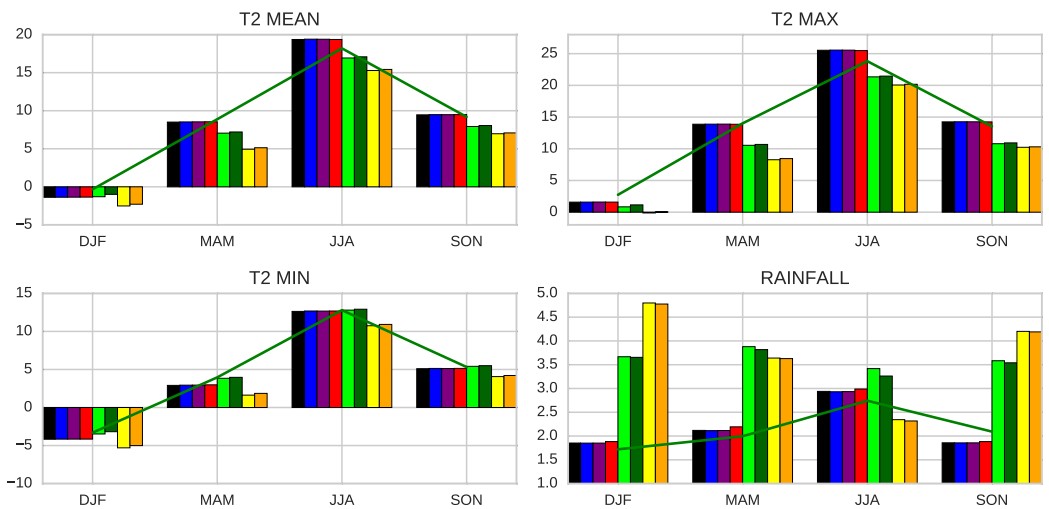

**Figure 2.** Averaged seasonal modelled temperatures (in °C) and precipitation (mm) for individual model simulations: Black – WRF–woU, blue – WRF–BULK, purple – WRF–SLUCM, red – WRF–BEP+BEM, light green – RegCM–SLUCM–woU, dark green – RegCM–SLUCM, yellow – RegCM–CLM4.5–woU, orange – RegCM–CLM4.5. Green curve stand for the seasonal values given by E-OBS.

Another important question is how are the models able to predict the temperature difference between urban area and its non-urban vicinity. To evaluate this, we chose three big cities, Berlin, Munich and Prague, as they are located inland without significant orographic variation that would mask the urban canopy effects. We took one station from each city center (Berlin–Mitte, Munich–Bogenhausen and Prague–Karlov) and one from its surroundings (Berlin–Schönefeld, Munich–Flughafen and Prague–Ruzyně). To prevent the impact of different altitudes of the listed stations and grid-points, adjustment was performed

using standard temperature moist adiabatic gradient of 0.65 K/100 m. As written above, the urban versus rural temperature difference can vary substantially depending on the specific weather conditions, time of the day and year. We therefore focus on the distribution of these differences (Fig. 4). The range between the 5th and the 95th percentile is mostly well captured, but often overestimated in the simulation using the most sophisticated urban model WRF–BEP+BEM. Also, the distribution





**Figure 3.** Spatial distribution of averaged daily precipitation biases (mm) for individual seasons and simulations. The white color indicates missing reference data.





of differences is often substantially different for the W–BB simulation compared to the rest of the analysis setups. On the other hand, in terms of summer temperature minima, all WRF simulations give distributions with the median significantly shifted to higher values.

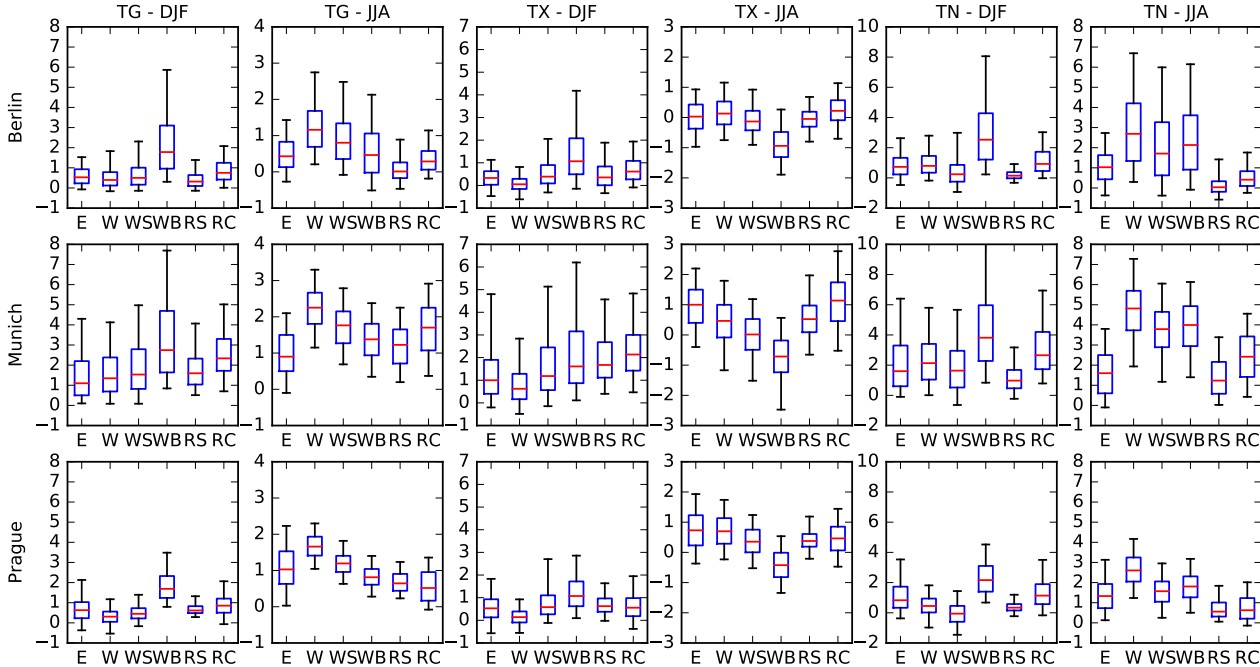

**Figure 4.** Distributions of temperature differences (°C) between city centres and their surroundings in winter and summer season given by station data (E), WRF–BULK (W), WRF–SLUCM (WS), WRF–BEP+BEM (WB), RegCM–SLUCM (RS) and RegCM–CLM4.5 (RC) model simulation. $TG$ denotes temperature means, similarly $TX$ and $TN$ stand for maxima and minima, respectively. Whiskers indicate 5th and 95th percentile.

Over Prague, hourly temperature data are available for both the city center and from a station in the surrounding area, which enables a detailed comparison of temperature daily cycles. Again, the adjustment of temperature to the sea level was performed using the standard temperature lapse rate. Results are shown in Fig. 5. Modelled values are often significantly biased (e.g. winter minima and summer maxima by WRF, daily means by RegCM in spring, summer and autumn). Furthermore, the diurnal temperature range is evidently overestimated by the WRF model with the exception of winter and underestimated by the RegCM model, which is consistent with the domain averaged temperature biases (Fig. 2).

We are also interested in temperature differences between the city center and its surroundings. From this perspective, models are qualitatively able to capture the evening- and night-time city center temperature increase in warmer seasons. While the RegCM model tends to underestimate this effect, WRF rather overestimates it in all simulations. Nearly all simulations show





a zero temperature difference for the morning hours, in line with the station data. However, the W–BB simulation manifests the UCI in the morning, occasionally observed during this part of the day in Prague, but not on average over each day. In winter season, the W–BB simulation also overestimates the temperature difference during the whole day. From the perspective of temperature differences, we can conclude that all urban schemes give qualitatively good results and there is no significant

5   improvement from using the more sophisticated urban model, in comparison to the simple BULK scheme.

Temperature in Prague city center (red), vicinity (green), models (solid), ref. (dotted)

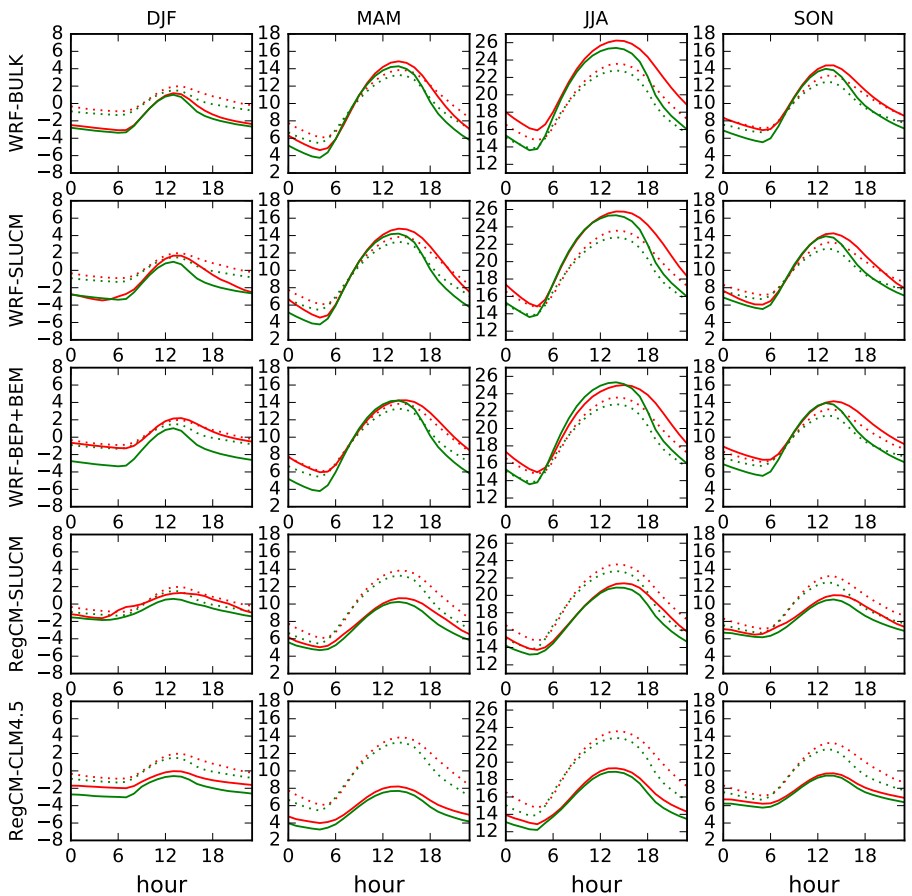

**Figure 5.** Averaged temperature daily cycles (°C) in the Prague city center (station Karlov, red) and its surroundings (station Ruzyně, green). The comparison of modelled (solid) and measured values (dotted). The time axis corresponds to UTC.





## 3.2 Benefits of more complex approaches of urban parametrizations

In this section, we will show the long-term impact of urban surfaces on meteorological conditions both near the surface and at higher altitudes. This can give us an insight into the impacts of urban environment on selected immeasurable meteorological variables. At the same time, we will evaluate the benefits and/or disadvantages of the specific model approaches. First, the impact of urban surface to the temperature is shown (Fig. 6). Temperature daily cycles pertain to the city of Prague and its surroundings. More specifically, the city center value is taken from a grid-point with a position closest to the actual city center and containing the urban land-use. The surroundings are defined as a square ring of grid-points with a distance of two grid-points around the central grid-point. For comparison, values from the simulations with urban land-use removed are also taken into consideration, in this case from both the central grid-point and the square ring.

As stated in the previous section, all model approaches are able to capture evening and nocturnal UHI in the warm season, although the magnitudes differ. The BULK model increases summer temperature maxima more than the urban canopy models, which is probably caused by disregarding the three-dimensional character of urban areas. The winter temperature cycles differ more between each other, which is consistent with the substantial differences in the implementation of the Anthropogenic Heat production. The differences between the temperature profiles and profiles given by non-urban simulations over city surrounding are very low, indicating that the impact of the city on its surroundings over a distance of a few tens of kilometres is minor.

The impact of urban surfaces on the height of the planetary boundary layer (PBLH) is plotted in Fig. 7. It could be expected that the PBLH will be increased by higher intensity of friction in the urban environment and by increased buoyancy due to temperature increase. Model results indeed show PBLH increase during the whole day except morning hours, when temperature increases are minimal, too. In winter, model results differ much more than in the rest of the year, in agreement with the temperature profiles. Again, impact of the city to its surroundings is mostly negligible, though in winter in the RegCM simulations, the value is about a few tens of meters. There is a significant difference in the PBLH between the two models; e.g. the PBLH is about 25 % lower in the RegCM model in comparison to the WRF model in summer.

In terms of the urban surface impact on wind speed (Fig. 8), the results show significant differences between individual models and urban schemes. Only a minor impact is detected in the WRF–BULK simulation, while much higher impacts are detected in the WRF–SLUCM and WRF–BEP+BEM simulations. On the other hand, impacts in the RegCM simulations are low (especially in summer), despite of using the urban canopy model. There is a clearly visible wind speed increase during daytime in all WRF simulation, caused probably by thermal processes and convection. In the evening, when the UHI manifests, these processes have higher intensity in cities than in their surroundings. During the evening, the differences in wind speed between city centres and their surroundings are also lower. In the RegCM simulations, summer wind speed daily amplitude is rather small with hardly visible hourly variation.

Similarly to their effects on surface variables, the impacts of urban surfaces on vertical profiles of meteorological variables are computed from the difference between simulations with and without urban land-use. In Fig. 9, the impact of the urban surfaces on vertical distribution of temperature is shown over three selected cities (Berlin, Munich and Prague). In most of cases, this impact is visible up to the altitude of approximately 1 km in winter, and 2 km in summer. In general, the urban



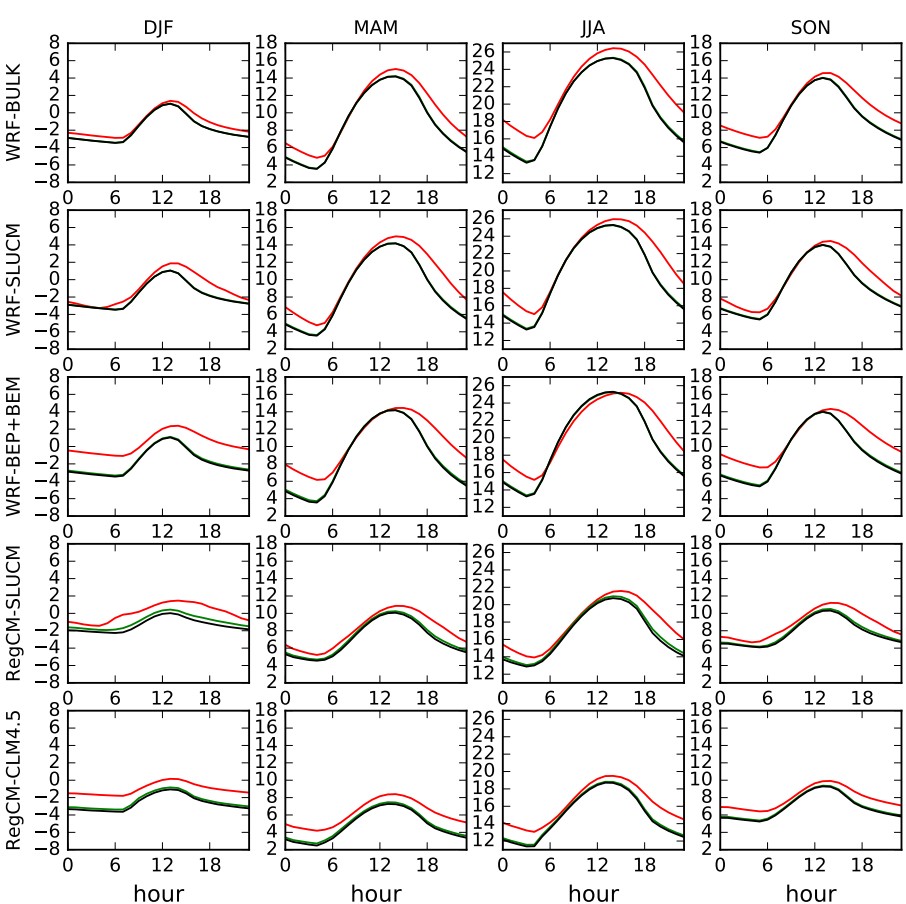

**Figure 6.** Averaged temperature daily cycles (°C) in the Prague city center (red), in its surroundings (green) and those simulated with the urban surface removed (black).



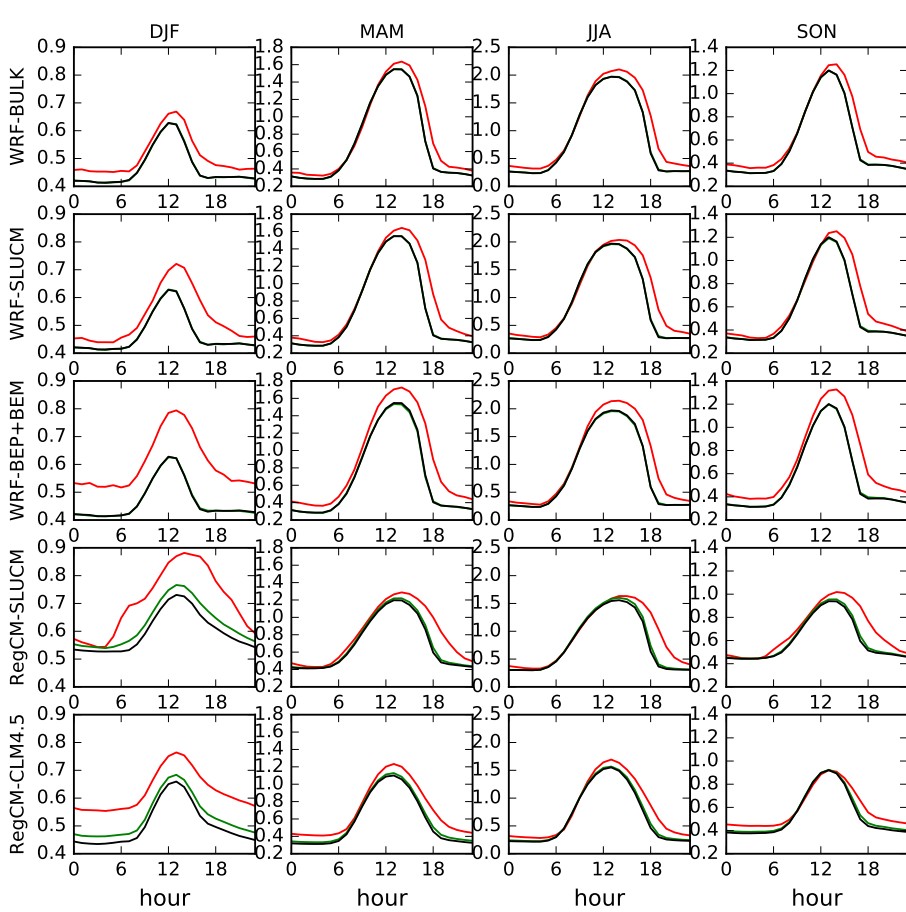

**Figure 7.** Averaged daily cycles of the planetary boundary layer height (PBLH) in the Prague city center (red), in its surroundings (green) and as simulated with the urban surface removed (black) in kilometres.





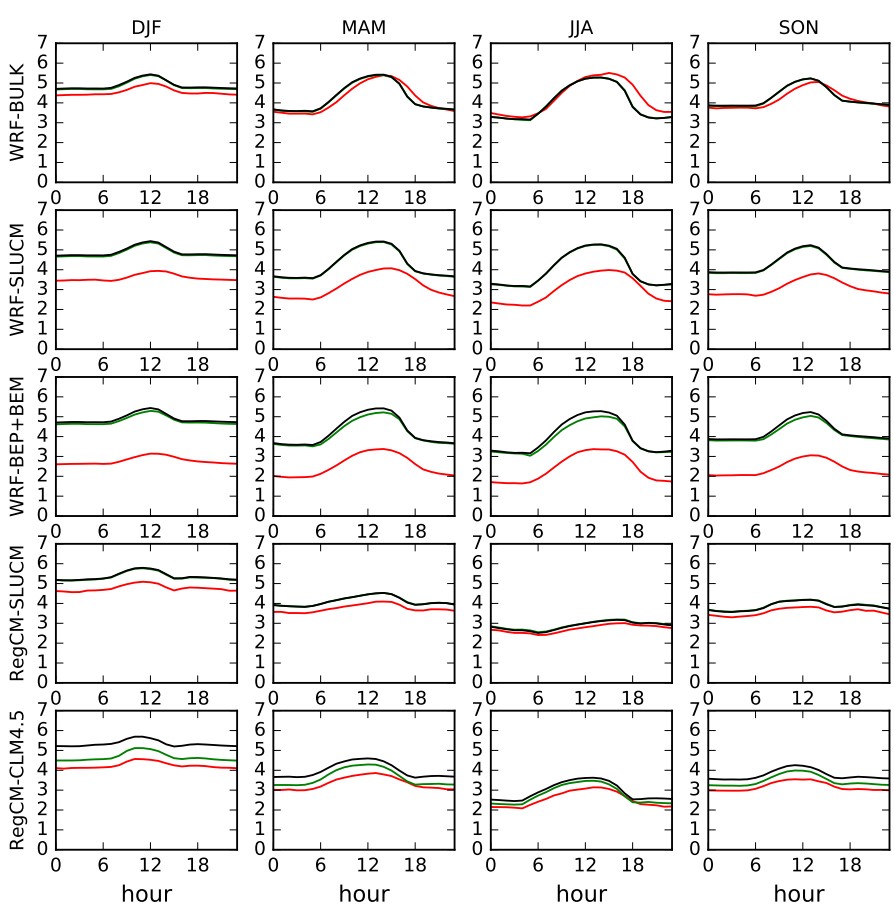

**Figure 8.** Averaged horizontal wind speed daily cycles ($\mathrm{m\,s^{-1}}$) in the Prague city center (red), in its surroundings (green) and as simulated with the urban surface removed (black).




effects in simulations with RegCM propagate to higher altitudes, however, the magnitudes close to the surface are usually smaller than in WRF. In winter, they are mostly between 1–2 °C in average. The W–BB simulations seems to be the warmest over cities but as said, over higher altitudes, the impact in RegCM is higher. In summer, the spread between models and urban parametrizations is smaller, mainly due to the fact that anthropogenic heat (which is not included in all experiments) plays a smaller role here. During this season, the effects in WRF sharply become small over the altitude of 200 meters, while in RegCM (in the R–S) experiment, urban induced temperature warming can be as much as 0.5 °C at such altitudes.

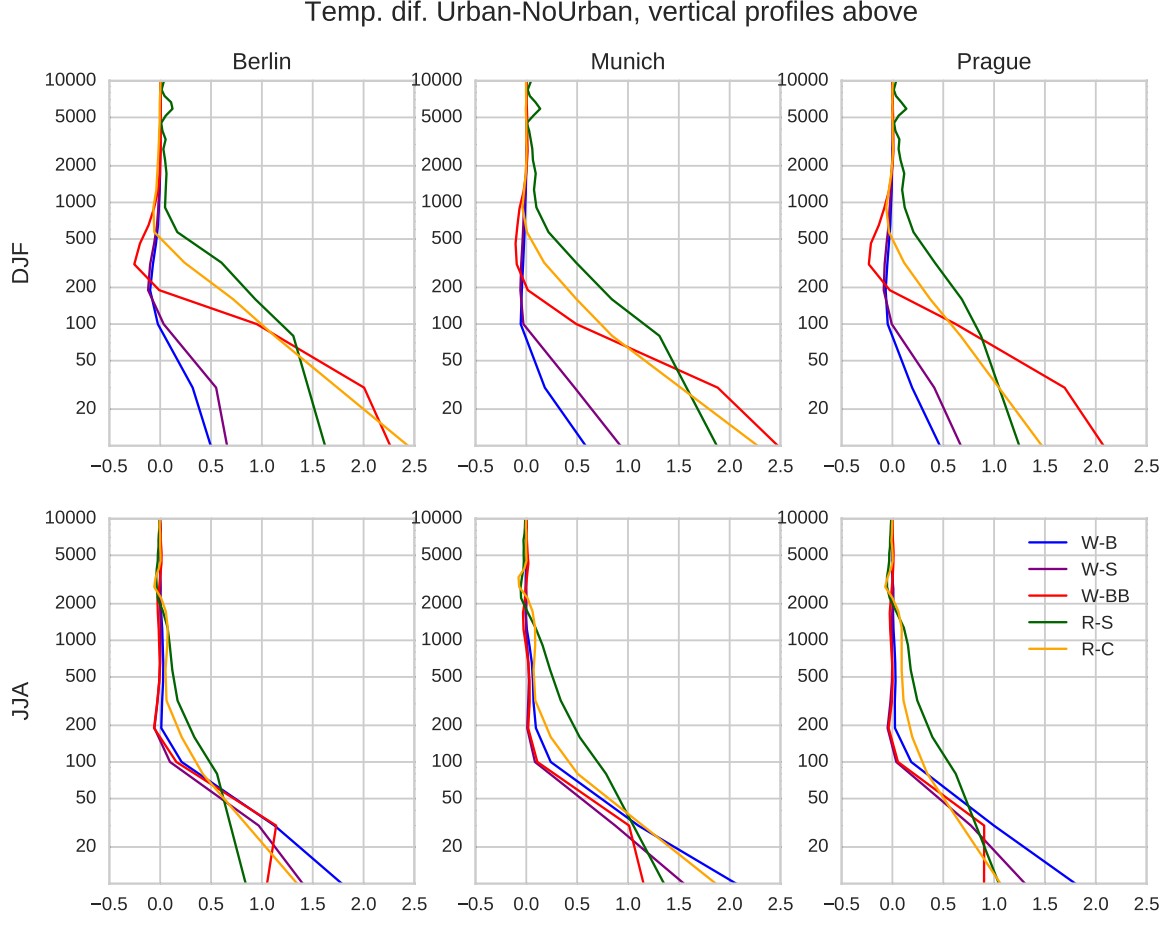

**Figure 9.** The impact of the urban surface inclusion on the temperature vertical profile (in °C) over selected cities. The Y–axis indicates the altitude in meters.

Fig. 10 shows the impact of the urban surfaces on the vertical profile of the horizontal wind speed for Berlin, Munich and Prague. Again, the impact is distinct up to 1 or 2 km and the highest impact is simulated in the WRF–BEP+BEM experiment (decrease of the wind speed over 2 m s$^{-1}$), where the impact increases with height in the first model layers. The remaining





WRF simulations indicate less wind speed decrease in the overall profile, about $1.5 \, \mathrm{m \, s^{-1}}$ in the W–S and about $0.5 \, \mathrm{m \, s^{-1}}$ in the W–B simulation in winter; and about $1 \, \mathrm{m \, s^{-1}}$ in the W–S and up to $0.5 \, \mathrm{m \, s^{-1}}$ (but with a small wind speed increase at the surface up to $0.3 \, \mathrm{m \, s^{-1}}$) in the W–B simulation in summer, respectively.

5     The RegCM simulations, in general, generate smaller impacts compared to WRF. The CLMU scheme gives an impact between 0.5 and $1.5 \, \mathrm{m \, s^{-1}}$. The RegCM simulation with the SLUCM scheme shows, however, a very small surface wind speed reduction (up to $0.5 \, \mathrm{m \, s^{-1}}$).

**Figure 10.** The impact of the urban surface inclusion on the wind speed vertical profile ($\mathrm{m \, s^{-1}}$) over selected cities. The Y–axis indicates the altitude in meters.



### 3.3 Consequences to pollutant dispersion

Although this study is not investigating the impact of the urban surfaces on air quality, e.g. by using a coupled meteorological and chemical-transport model, we would like to outline a basic concept of how the urban surfaces impact the air quality due to modified weather and climate conditions. To achieve this, we will evaluate impacts of the urban surfaces on variables which describe the ability of the atmosphere to disperse pollutant plumes. First, this ability is given by the cleaning effect of the wind in the boundary layer, secondly by convection and thermally inducted turbulence. To quantify these effects, we chose two indexes used by the Czech Hydro-Meteorological Institute (CHMI). The cleaning effect of the wind is described by the Ventilation Index (VI; Hardy et al., 2001):

$$VI = PBLH \cdot \overline{v}, \tag{1}$$

where $\overline{v}$ is the mean (in vertical) wind speed within the boundary layer. The dispersion effect of convection and turbulence is quantified by the Stability Index (SI; Bubník et al., 1998), which indicates the vertical stability of the atmosphere:

$$SI = \frac{T_{2m} - T_{850\,hPa}}{H_{850\,hPa}}, \tag{2}$$

where $T_{2m}$ is the ground temperature, $T_{850\,hPa}$ the temperature at the 850 hPa level and $H_{850\,hPa}$ the mean altitude of the 850 hPa level.

Due to the fact that the wind speed is decreased by cities and conversely the PBLH is increased, it is not possible to simply infer the impact of cities on the VI. Its magnitude depends on the specific weather situation and the time of eventual coincidence between small or high values of the relevant variables. The time series of the VI does not follow the Gaussian distribution, so mean values may not be sufficiently representative and an entire distribution of the VI has to be shown. Seasonal distributions of the VI for the city of Prague are displayed in Fig. 11. In winter, there is a significant difference between the VI distributions given by the WRF and RegCM simulations. The RegCM model gives much flatter distributions and thus more favourable from the perspective of the pollutant dispersion. However, despite this great difference, all simulations indicate that the urban inclusion has a small positive impact on the VI. In summer, the differences between the WRF and the RegCM are smaller. Here, the inclusion of urban canopy effects increases the VI too leading again to more favourable dispersion conditions.

The seasonal distributions of the SI for the city of Prague are shown in Fig. 12. The negative values of the SI indicate the temperature inversion, i.e. a highly stable state of the atmosphere, with adverse conditions for the pollutant dispersion. Higher values of the SI imply more dispersive conditions. Values of the SI greater than $1 \cdot 10^{-2}\,\mathrm{K\,m^{-1}}$ indicate unstable state, the best from the perspective of pollutant dispersion. The winter SI distributions show very sharp local maximum for zero SI, which is probably related with frequent occurrence of elevated temperature inversions. The position of the second maximum depends on the specific simulation. Generally, simulations with the urban surfaces shift the maximum towards higher values, indicating a positive impact of urban surfaces on the winter SI. In summer, a local maximum around zero also appears. During the summer nights, the inclusion of the urban surfaces increases the value of the SI, but in the daytime, the effect is only minor or negative (for the W–BB simulation). In summary, the urban surfaces slightly increase the SI and improve the meteorological conditions to favour pollutant dispersion.





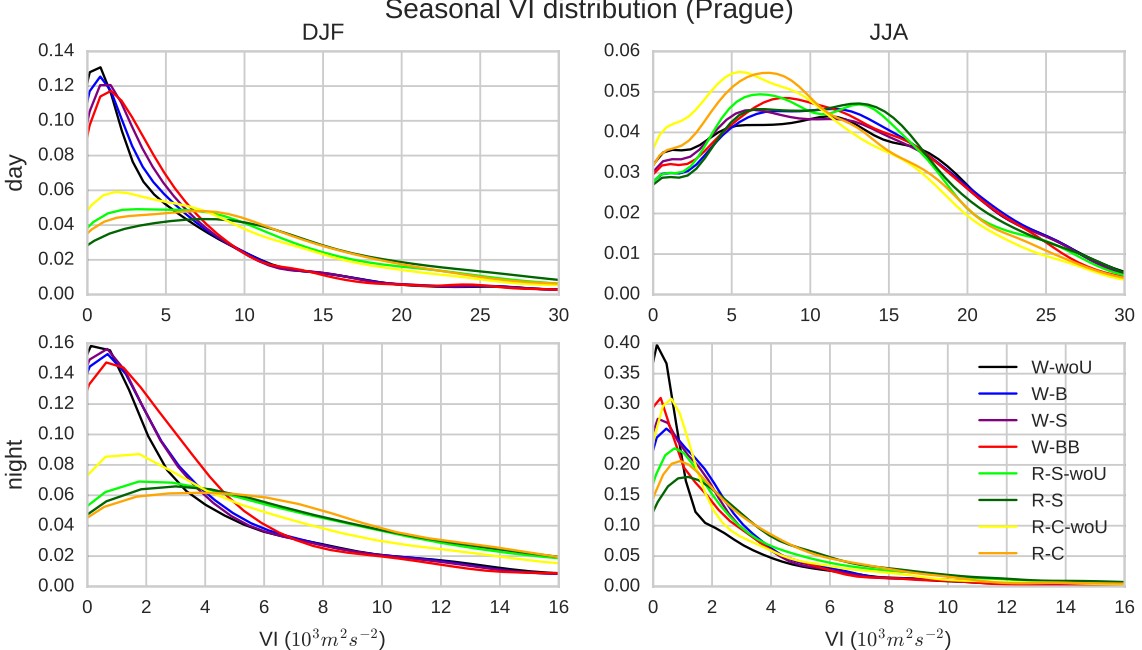

**Figure 11.** Seasonal distributions of the Ventilation Index (VI; in $10^3 m^2 s^{-2}$) for the city of Prague, given by all simulations. The day represents the time between 11–16 UTC, the night represents the time between 23–04 UTC.

## 4 Discussion

The WRF simulations are characterized by the mean temperature biases lower than 1 °C (Fig. 2), making them a reasonable basis for the analysis of the effects of urban surfaces. Mar et al. (2016) using WRF model over Europe reached lower temperature biases for their domain, but these values are highly dependent on the chosen physical parametrizations, domain parameters

and the area of interest, as described by Katragkou et al. (2015). This is true for both temperature and precipitation biases. The RegCM simulations with the BATS land-surface scheme give temperature biases that are comparable with Huszar et al. (2016), who also employed the RegCM model with the BATS scheme. The highest temperature biases from all simulations were detected for the RegCM model with the CLM4.5 scheme. The precipitation is captured much better by the WRF model than by the RegCM model, which exhibits high precipitation overestimation. The annual cycle given by the RegCM simulation

with the BATS is distinctly different from the cycle presented by Huszár et al. (2016), with the precipitation maxima occurring in winter instead of summer, given by Huszár et al. (2016) as well as the observational E-OBS data. For the CLM4.5 scheme, the precipitation characteristics are similar in shape, with strongly expressed biases. Torma et al. (2011) and Zanis et al. (2015), using an earlier version of the RegCM and performing experiments with similar resolution, concluded that the Grell convective scheme tends to overestimate precipitation over mountainous regions. Another contributing factor can be the too wet model

atmosphere caused by increased evaporation, which is indeed observed in the BATS scheme (Winter et al., 2009). Thirdly,



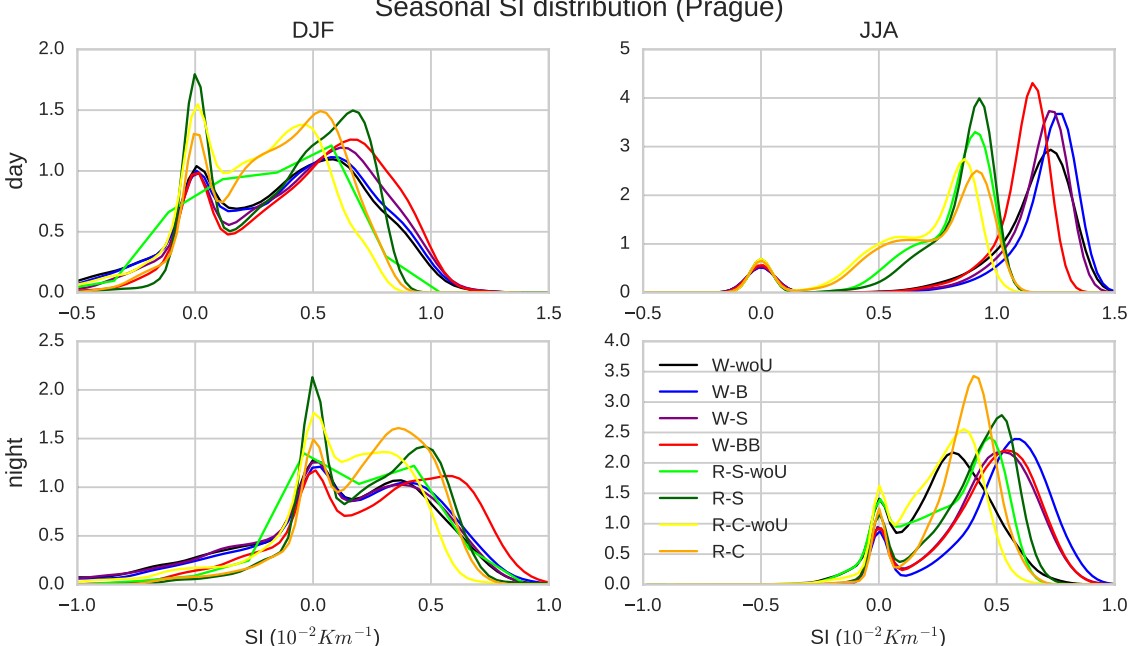

**Figure 12.** Seasonal distributions of the Stability Index (SI; in $10^{-2}\,\mathrm{K\,m^{-1}}$) for the city of Prague, given by all simulations. The day represents the time between 11–16 UTC, the night represents the time between 23–04 UTC.

overestimation of precipitation can be attributed to the autoconversion and raindrop evaporation rate – these parameters were tuned by Torma et al. (2011), who tried to reduce the precipitation bias in the RegCM. However, even with this tuning some positive precipitation still remained. The observed temperature bias links to the precipitation bias: too wet model atmosphere leading in increased cloudiness and/or precipitation resulting in lower insolation reducing the surface temperatures.

The histograms of temperature differences between city centres and their surroundings (Fig. 4) show that models are able to capture the different intensity of the UHI, depending on the specific weather conditions. Model ranges of temperature differences are often even higher compared to the reference data. The urban parameters (Table 1) were chosen to describe the urban environment corresponding to the city of Prague. This can be one of the reasons why the UHI was captured with the highest agreement for this city and this stresses the importance of accurately setting urban canopy parameters. While Sarkar

and De Ridder (2011) showed that the summer UHI intensity is underestimated by simple urban model in the city of Paris (by 0.5 °C), an equivalent WRF–BULK simulation indicates clear overestimation of the averaged summer UHI intensity. Trusilova et al. (2016) and Sharma et al. (2017) arrived to the same conclusion that suggests our results. Our results are well in line with the urban canopy induced temperature increases modelled by Trusilova et al. (2008) for central Europe, who used a single-layer urban canopy model, although our WRF–SLUCM simulation, which is closest to their setup, shows rather overestimation. The

W–BB simulation has a slightly different character in this regard, exhibiting over-sensitivity to the urban canopy forcing as it reacts to urban surface with a much stronger increase of the winter temperature, the summer UCI and winter minima compared



to other models. This feature can be partly seen in Liao et al. (2014), where the BEP+BEM also gives the highest summer temperature minima and increases the winter temperature mean. However, in their study, which focused on the Chinese urban environment, the BEP+BEM scheme gives daily temperature profiles that are the closest ones to the reference data. This may indicate that input data can be one of the reasons why this urban canopy model performs less accurately. Further, the too strong

response in the BEP+BEM setup can also be caused by unfulfilled preconditions in the central European cities: e.g. no air conditioning but instead, using of window blinds.

In all simulations, negative winter temperature biases (Fig. 5) are consistent with biases in Fig. 4. This is also true for higher summer temperature maxima given by the WRF model, and for lower temperatures in spring, summer and autumn given by the RegCM model, especially with the CLM4.5 scheme. The diurnal temperature range (DTR) is overestimated in the summer

season similarly to Liao et al. (2014). In the RegCM model, the DTR is underestimated, contrary to Huszar et al. (2014), where the DTR is rather overestimated. In terms of differences between the city and the surrounding areas, there is a good agreement in the summer morning minimum of temperature difference, except for the W–BB simulation, in which due to the over-sensitivity of the urban effects, a strong UCI occurs. The lower intensity of the evening and nocturnal summer temperature increase given by the RegCM model is very similar to Huszar et al. (2014).

In general, simulations are able to capture the summer evening and nocturnal UHI (Fig. 5 and Fig. 6), with somewhat lower intensity of this phenomenon in the RegCM simulations. The BULK model gives the highest summer temperature maximum, which is in agreement with Liao et al. (2014). Also the evening and nocturnal UHI is the most intense in the WRF–BULK simulation, same as observed in Sharma et al. (2017), who also compared urban parametrizations in the WRF model for summer episode for different urban types. In winter, there is a more notable difference between specific simulations. The

simple BULK scheme gives only a small impact of the urban environment (Fig. 6), in comparison to other simulations with urban canopy schemes. This is probably caused by disregarding anthropogenic heat. Regarding the temperatures, advantages of more complicated urban environment schemes seem to be only minor, perhaps the most evident improvement occurs with the winter UHI. Our results rather indicate that the most sophisticated scheme (BEP+BEM) often over-emphasizes the urban effects, e.g. the summer morning UCI, which can only appear if specific weather conditions occur (Theeuwes et al., 2015), or

the increased winter UHI. This conclusion contrasts with the results of Trusilova et al. (2016), where the multi-layer scheme is the only scheme that is able to correctly capture the summer UHI daily cycle. All model simulations produce the temperature maximum time-shift about 1–2 hours for the Prague city center, in all seasons. This is in agreement with idealized temperature profiles for urban and rural areas given by Oke (1982) and also with observed values for the Prague area, showed by Huszar et al. (2014) or partly by Fig. 5 (besides summer).

In terms of the urban environment impact on the planetary boundary layer height, the differences between specific simulations are more apparent. As expected from the underlying physics and as shown in Fig. 7, the PBLH urban increase is generated by higher intensity of friction and by increased turbulent mixing caused by temperature increase. In winter, despite the low intensity of UHI, the PBLH is increased mainly by greater friction. Conversely, in the summer mornings, when the UHI is zero, there is no PBLH increase in any simulation, because the absolute value of the PBLH is too high to be impacted by friction.

The BEP+BEM simulation gives the greatest increase of the PBLH, which is consistent with Liao et al. (2014).



The impact of the urban surfaces on the wind speed in 10 m is also highly dependent on the used urban scheme and the model (Fig. 8). Only a low negative impact, mostly less than 1 m s$^{-1}$, is indicated in all RegCM simulations. This is consistent with the results by Huszar et al. (2014) and also by Huszár et al. (2018), with smaller impacts in summer than in winter. Except for the BULK scheme, the WRF model gives a greater wind speed reduction for the city (between 1 and 2 m s$^{-1}$), for all

seasons and times of day. The reduction is found to be the highest in the WRF–BEP+BEM simulation, which is consistent with Liao et al. (2014), despite smaller differences between the W–S and the W–BB simulations in our experiments. Regarding the wind speeds, there are significant differences between profiles given by specific urban schemes, stressing the high uncertainty of parametrization of wind within urban environment.

Similarly to impacts of urban surfaces on the 2 m temperature, the WRF model with the BULK and SLUCM scheme pro-

duces only small changes in overall temperature profile, in winter (Fig. 9). The W–BB simulation and both RegCM simulations alter the profile more significantly. In summer, the impacts in individual simulations are much more similar to each other. A weak negative impact at higher altitudes, between 2 and 10 km, indicated by Huszar et al. (2014) in their RegCM simulation, also appears, but the magnitude is negligible. As same as in Fig. 4, the overestimated average winter UHI at low levels (caused probably by unfulfilled preconditions for the AH flux computation) in WRF–BEP+BEM simulation leads to the opposite an-

nual cycle of the UHI intensity in the Prague, which contrasts with observations. The UHI annual cycle is also opposite for the both RegCM simulations, which is caused by slightly overestimated winter UHI and slightly underestimated summer UHI. In RegCM, the AH was prescribed as annual mean and monthly disaggregation factors, which decompose this annual value into monthly values. From the results, we can conclude that the winter AH release is probably too high leading to the UHI overestimation in winter.

Maybe it is the impact of urban surfaces on the wind speed profile (Fig. 10) where individual models/parametrizations differ the most, and not only in the shape of the profile but also in the magnitude both near the surface and at higher elevations up to a few hundreds of meters. Correct simulation of wind profiles and the whole structure of the PBL in urban environment is crucial for air-quality oriented studies and our results indicate that large uncertainty lies in the selection of the model-UCM pair.

In terms of the impact of the urban surface on the ventilation index (Fig. 11), there is one important question: why are the winter distributions given by the RegCM model flatter than distributions given by the WRF model? One reason is that the negative impact of the urban surface on the surface wind speed is much weaker in the RegCM model and also the averaged RegCM surface wind speeds are higher in cities as well as in their surroundings (Fig. 8). The second reason is that the PBLH computed by the RegCM model is higher during the night, when weather conditions unfavourable to pollutant dispersion

usually occur. The RegCM–SLUCM simulation gives the highest winter PBLH during all times of day. The winter urban VI increase can be explained by the increased PBLH, which means that the averaged wind speed is computed from a thicker boundary layer, so the lower surface wind speed in the city impacts the VI only slightly, due to the general wind speed increase with the altitude and the low impact of cities on the wind speed profile (Fig. 10). In summer, the small increase of the urban VI is caused mainly by the PBLH urban increase, manifesting for only some parts of the day. However, during the night, when the

PBLH has the lowest value, the urban PBLH increase occurs and the urban VI is also increased. Despite the prevailing positive





urban surface impacts on the VI, the impact may be negative during some parts of the day. For instance, in morning hours, when the intensity of the UHI is zero, the urban PBLH is not increased, but the wind speed is decreased, leading to decreases in urban VI.

The impact of the urban environment on the SI (Fig. 12) shows several interesting features. A strong peak in the SI distri-
bution appears for zero SI in all seasons and daytimes. In winter, the frequent occurrence of elevated temperature inversions probably induces this maximum. In summer, rainy episodes are likely responsible, due to great thermal capacity of water, equalizing the surface and the 850 hPa temperature. Conversely, a windy episode should cause strong mixing; the SI then falls between the values of the adiabatic gradient for dry or saturated air, i.e. between 0.6 and $1 \cdot 10^{-2} \mathrm{K\,m^{-1}}$. It is therefore not easy to explain the distribution maximum during the summer days given by all WRF simulations that occurs for the SI greater than
$1 \cdot 10^{-2} \mathrm{K\,m^{-1}}$ (strongly convective conditions). Perhaps the near surface temperature increase in the afternoon could cause this strong temperature gradient. It is not evident why the R–S–woU simulation does not create the zero SI peak in winter.

The increase of the VI and SI in the city and improved weather conditions with regard to the pollutant dispersion are consistent with the results by Liao et al. (2014), who showed a decrease of PM10 species concentration in simulations with the BEP and BEM schemes against the single layer urban canopy scheme, in both winter and summer. Similarly, Huszár
et al. (2018) described a small NO and $NO_2$ decrease induced by the urban environment. Conversely, the $O_3$ concentration increases in the city, due to better mixing of primary pollutants caused by higher boundary layer and by greater intensity of the turbulence in the urban environment. We can assume that the urban environment impact on the surrounding rural VI and SI will be negligible, because the impact of cities on the rural temperature, PBLH and wind speed is also mostly negligible.

## 5 Conclusions

We performed series of simulations with the WRF and RegCM models, together with different descriptions of the urban environment. The simulations were focused on the area of the central Europe, with 10 km horizontal resolution, over the 2001–2010 period. Effects of cities were detected from differences between results of simulations with and without urban surfaces included. The validation of urban impacts on the temperature was performed using the station data from the ECAD and from the Czech Hydro-Meteorological Institute.

Our study shows that in the long-term averages, all urban schemes are able to capture the main urban meteorological feature, the evening and night-time city temperature increase (called UHI), which mainly occurs in the summer season with the intensity of 2–3 °C on average. Moreover, all models also reproduce the morning temperature equality of the city and its surroundings. From the perspective of the temperature, there is no significant improvement of model outcomes stemming from the use of a more complicated urban scheme. This is true not only for averaged daily temperature profiles in the city against its
surroundings, but also for entire distributions of their differences, including daily extremes. In winter, the city temperature increase is significantly affected by the anthropogenic heat computation process.

The impact of the urban environment on the PBLH and on the surface wind speed is more dependent on the used model and urban parametrization. For example, during summer noon, the PBLH is about 25 % lower in RegCM simulations than in




the WRF simulations. In terms of the wind speed, the impact of the city is lower in the RegCM simulations (reduction of only about 1 m s$^{-1}$), in comparison to the WRF simulations with the urban canopy scheme (reduction of 1–2 m s$^{-1}$). Temperature and wind speed profiles are impacted by cities up to approximately 2 km, with mostly decreasing tendency from the surface values to the zero. The values of the urban induced impacts in the lowest model layers mostly follow up the values of these

impacts at the surface. Due to the absence of reference data for these poorly observed variables, we are unable to assess which model setup gives the best results. However, our results are mostly consistent with the outcomes of previous studies, and should therefore provide a reasonable basis for the study of the effects of urban surfaces.

The evaluation of the indices describing meteorological conditions from the perspective of the pollutant dispersion showed that the urban environment slightly improves these conditions in comparison to the non-urban setup, except during the summer

days. This is true for both the cleansing effect of the wind and the dispersion effect of convection and turbulence. However, it must be stressed that only the concentration of primary pollutants can be decreased by urban effects (e.g. particulate matter, sulphur and nitride oxides), not the secondary pollutants that are created in the atmosphere (e.g. ozone), as shown by Huszár et al. (2018).

*Code and data availability.* The source code of the WRF model is publicly available (after registration) on http://www2.mmm.ucar.edu/wrf

/users/download/get_source.html. The modelled data used in this study can be provided upon request to the corresponding author.

*Competing interests.* The authors declare that they have no conflict of interest.

*Acknowledgements.* This work has been funded by the project PROGRES – Programme of Charles University Q16, by the project OP-PPR Proof of Concept, No. CZ.07.1.02/0.0/0.0/16_023/0000108 and by Charles University project SVV No. 260327. Authors wish to express their thanks to WRF and RegCM team communities for their development of models and urban modules used in the study. We acknowledge the

E-OBS dataset from the EU-FP6 project ENSEMBLES (http://ensembles-eu.metoffice.com) and the data providers in the ECA&D project (http://www.ecad.eu). This work was supported also by The Ministry of Education, Youth and Sports from the Large Infrastructures for Research, Experimental Development and Innovations project „IT4Innovations National Supercomputing Center – LM2015070".





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
