# Peer review of "Multi-model comparison of urban heat island modelling approaches"

_Atmospheric Chemistry and Physics, 2018_

## Referee Comment (RC1) · Anonymous Referee #1 · 12 May 2018

This is an interesting paper investigating the role of urban surface in regional climate in central Europe via using WRF and RegCM model simulation. The paper is generally well-written and the conclusions are adequately supported by the evidence presented. I have following comments before it is accepted to be published on ACP.

1. Page 4, Line 23 - 10km * 10km horizontal resolution and 30/23 vertical levels is a relatively low for a study of urbanization impact. Previous numerical studies generally used a higher resolution of less than 3km in order to better illustrate the region of urbanized area (e. g. Lin et al., 2008; Holt et al., 2009; Wang et al. 2012). I suppose this number is restricted by the used computer power, but could you add a comment about it? How the resolution number can influence the results?

Holt T, Pullen J, Bishop C. Urban and ocean ensembles for improved meteorological

and dispersion modelling of the coastal zone. Tellus Series A-dynamic Meteorology & Oceanography, 2010, 61(2):232-249.

Wang J, Feng J, Yan Z, et al. Nested high-resolution modeling of the impact of urbanization on regional climate in three vast urban agglomerations in China. Journal of Geophysical Research Atmospheres, 2012, 117(D21):-.

Lin C Y, Chen F, Huang J C, et al. Urban heat island effect and its impact on boundary layer development and land–sea circulation over northern Taiwan. Atmospheric Environment, 2008, 42(22): 5635-5649.

2. Figure 2 - I suggest the author add a color bar (and the corresponding model) on the right side of the figure instead of the description in figure caption to make it easier to understand.

3. Figure 5 - Why the temperature is generally overestimated by WRF but underestimated by RegCM model? Can you give some possible explanations for it?

4. Figure 11- what does the Y-coordinate represent for Figure 11 and 12? Does it mean the frequency distribution of VI and SI? The Y-axis title needs to be added.

5. Since the paper discussed the urbanization effect on pollutant dispersion, some important publications need to be mentioned in the manuscript.

Liao, J., T. Wang, Z. Jiang, B. Zhuang, M. Xie, C. Yin, X. Wang, J. Zhu, Y. Fu, and Y. Zhang (2015), WRF/Chem modeling of the impacts of urban expansion on regional climate and air pollutants in Yangtze River Delta, China, Atmospheric Environment, 106, 204-214, doi:10.1016/j.atmosenv.2015.01.059.

Tao, W., J. Liu, G.A. Ban-Weiss, D.A. Hauglustaine, L. Zhang, Q. Zhang, Y. Cheng, Y. Yu, and S. Tao (2015), Effects of urban land expansion on the regional meteorology and air quality of eastern China, Atmospheric Chemistry and Physics, 15, 8597-8614.

[Figure]

2018.

---

## Referee Comment (RC2) · Anonymous Referee #3 · 17 May 2018

In this manuscript, the authors have investigated the role of urbanized surfaces on micrometeorology and air dispersion potential over cities in central Europe. To address this, the paper has used a series of decade long regional simulations with various available urban modules in WRF and RegCM model. This is a well-written paper with clear results and conclusions. This research paper has considerable scientific significance. However, the authors should address the following points, before it is accepted to be published on ACP.

1) Figure 1 : One more panel illustrating the land use land cover mapping at fine (∼1 km or so) resolution will help to compare and understand how well the cities have been represented in the 10 km resolved simulations. Also the location of the cities used in analysis should be marked for ease of readers.

[Figure]

2) Figure 9 and 10: Large uncertainties exist in urban-induced differences in vertical profiles of temperature and wind speed when compared across various configurations. But as also concluded by authors, lack of evaluation limits the ranking of the configurations used. In this regards, evaluation of the vertical profiles of temperature and wind speed against radiosonde observations (over or near these cities) could facilitate improvement in conclusions. Radiosonde observations over the domain of study are available openly from http://weather.uwyo.edu/upperair/sounding.html.

3) Figure 11 and 12: Showing the percentage change in SI and VI due to urban surface will better underline the significant of the urban-induced differences in VI and SI.

4) The conclusion that urban-induced modification enhances pollution dispersion is mainly based on the analysis over Prague. Authors should also check over other big cities in Europe to illustrate robustness of this association.

5) Many short-term urban sensitivity simulations around the globe have shown that urban surfaces enhance convergence of low level horizontal wind over city center (Shepherd et al., 2005; Lin et al., 2008; Sarangi et al., 2018; Niyogi et al., 2017; Zhong et al., 2017). This process can enhance the advection of particulate matter towards city center. Please include analysis/discussion about relative changes in convergence compared to VI and SI for these decade scale runs.

6) Also, the impact of urban surfaces on vertical velocity should be analysed/discussed in context to the urban-induced changes in VI simulated.

Shepherd, J. M. (2005). A review of curr ent investigations of urban-induced rainfall and recommendations for the future. Earth Interactions,9(12), 1–27. https://doi.org/10.1175/EI156.1

Lin, C.-Y., F. Chen, J Huang, Y. A. Liou, W.C. Chen, W.N. Chen, and Shaw C. Liu, 2008: Urban heat island effect and its impact on boundary layer development and land-sea circulation over Northern Taiwan, Atmos. Environ., 42,5639-5649

Sarangi, C., Tripathi, S. N., Qian, Y.,Kumar, S., & Ruby Leung, L. (2018).Aerosol and urban land use effect onrainfall around cities in Indo-GangeticBasin from observations and cloudresolving model simulations. Journal ofGeophysical Research: Atmospheres, 123,3645–3667. https://doi.org/10.1002/2017JD028004

Niyogi, D., Lei, M., Kishtawal, C., Schmid, P., & Shepherd, M. (2017). Urbanization impacts on the summer heavy rainfall climatology over theEastern United States. Earth Interactions, 21(5), 1–17. https://doi.org/10.1175/EI-D-15-0045.1

Zhong, S., Qian, Y., Zhao, C., Leung, R., & Yang, X.-Q. (2015). A case study of urbanization impact on summer precipitation in the greaterBeijing metropolitan area: Urban heat island versus aerosol effects. Journal of Geophysical Research: Atmospheres, 120, 10,903–10,914.https://doi.org/10.1002/2015JD023753

---

## Author Comment (AC1) · 6 Jun 2018

We would like to thank to Anonymous Referee #1 for all comments, suggestions and corrections in his review of our manuscript. We addressed all and our point-by-point responses including the modifications in the manuscript follow:

**Referee's Comment #1:** 1. Page 4, Line 23 - 10km * 10km horizontal resolution and 30/23 vertical levels is a relatively low for a study of urbanization impact. Previous numerical studies generally used a higher resolution of less than 3km in order to better illustrate the region of urbanized area (e. g. Lin et al., 2008; Holt et al., 2009; Wang et al. 2012). I suppose this number is restricted by the used computer power, but could you add a comment about it? How the resolution number can influence the results?

[Figure]

**Author's response:** Yes, all mentioned studies computed with the horizontal resolution of 1–3 km and 28–40 vertical layers, but the time-range they analysed is much less than that in our study. Moreover, these studies focused on one or a few specific urban areas, while our study tries to be more general, concerning on all urban areas within a regional domain and on a long-term model comparison and impacts of urban surfaces. As such, 10 km x 10 km horizontal resolution a somewhat reduced vertical level number was chosen as a compromise. Finally, it is shown in Huszar et al. (2014) using the same resolution as us that considering dominant landuse instead of sub-grid landuse (2 km) leads to very similar impact of urban surfaces on the temperatures for sufficiently large cities (that we analyse in this study too).

**Referee's Comment #2:** Figure 2 - I suggest the author add a color bar (and the corresponding model) on the right side of the figure instead of the description in figure caption to make it easier to understand.

**Author's response:** We agree, the colour bar is added to the figure.

**Author's changes in manuscript:** The Fig. 2 is improved by adding a colour bar (attached to this reply).

**Referee's Comment #3:** Figure 5 - Why the temperature is generally overestimated by WRF but underestimated by RegCM model? Can you give some possible explanations for it?

**Author's response:** If we focus to the summer season, Fig. 2 shows that the main difference is in the temperature maxima. In the RegCM model, they are reduced by too wet model atmosphere and increased cloudiness, as written in the first paragraph of the discussion. In the WRF, increased temperature maxima can be explained by neglecting of realistic temporal and spatial distribution of aerosols in the model that use constant values of aerosol scattering properties, described also by Wang et al. (2012).

**Author's changes in manuscript:** A sentence about the temperature maxima overestimation in the WRF added into discussion to Figure 5: Maybe neglecting of the realistic temporal and spatial distribution of aerosols caused the summer temperature maxima overestimation, because the radiation scheme in the WRF uses values of scattering properties based on constant aerosol profiles, which is far from these in highly polluted urban areas.

**Referee's Comment #4:** Figure 11- what does the Y-coordinate represent for Figure 11 and 12? Does it mean the frequency distribution of VI and SI? The Y-axis title needs to be added.

**Author's response:** Yes, the Y-coordinate represents a frequency distribution, or, by other words, values of the estimated density function of the distribution. The area under all lines (or integral from density function) is equal to 1.

**Author's changes in manuscript:** Added information about the values of the estimated density function of the distribution to the caption of Figures 11 and 12.

**Referee's Comment #5:** Since the paper discussed the urbanization effect on pollutant dispersion, some important publications need to be mentioned in the manuscript.

**Author's response:** Yes, these studies also investigate the effects of the urban inclusion to pollution and the conclusions are very similar – the urban inclusion due to meteorological changes decreases near-surface PM10 (or other primary pollutants) concentrations and increases near-surface ozone concentration.

**Author's changes in manuscript:** Added sentence into the discussion: Also Liao et al. (2015) and Tao et al. (2015), who applied a coupled meteorological and chemical-transport model WRF-Chem with the SLUCM and BULK methods, found that the low level concentrations of primary pollutants decrease and ozone increase after the inclusion of urban surfaces.

Holt T, Pullen J, Bishop C. Urban and ocean ensembles for improved meteorological and dispersion modelling of the coastal zone. Tellus Series A-dynamic Meteorology Oceanography, 2010, 61(2):232-249.

Wang J, Feng J, Yan Z, et al. Nested high-resolution modeling of the impact of urbanization on regional climate in three vast urban agglomerations in China. Journal of Geophysical Research Atmospheres, 2012.

Lin C Y, Chen F, Huang J C, et al. Urban heat island effect and its impact on boundary layer development and land–sea circulation over northern Taiwan. Atmospheric Environment, 2008, 42(22): 5635-5649.

Huszar, P., Halenka, T., Belda, M., Zak, M., Sindelarova, K., and Miksovsky, J.: Regional climate model assessment of the urban landsurface forcing over central Europe, Atmospheric Chemistry and Physics, 14, 12 393–12 413, https://doi.org/10.5194/acp-14-12393-2014, https://www.atmos-chem-phys.net/14/12393/2014/, 2014.

Liao, J., T. Wang, Z. Jiang, B. Zhuang, M. Xie, C. Yin, X. Wang, J. Zhu, Y. Fu, and Y. Zhang (2015), WRF/Chem modeling of the impacts of urban expansion on regional climate and air pollutants in Yangtze River Delta, China, Atmospheric Environment, 106, 204-214, doi:10.1016/j.atmosenv.2015.01.059.

Tao, W., J. Liu, G.A. Ban-Weiss, D.A. Hauglustaine, L. Zhang, Q. Zhang, Y. Cheng, Y. Yu, and S. Tao (2015), Effects of urban land expansion on the regional meteorology and air quality of eastern China, Atmospheric Chemistry and Physics, 15, 8597-8614.

[Figure]

**Fig. 1.** Improved Fig. 2

---

## Author Comment (AC2) · 6 Jun 2018

We would like to thank to Anonymous Referee #3 for all comments, suggestions and corrections in his review of our manuscript. We addressed all and our point-by-point responses including the modifications in the manuscript follow:

**Referee's Comment #1:** Figure 1: One more panel illustrating the land use land cover mapping at fine (1 km or so) resolution will help to compare and understand how well the cities have been represented in the 10 km resolved simulations. Also the location of the cities used in analysis should be marked for ease of readers.

**Author's response:** One more figure similar to Fig. 1 mapping the terrain elevation,

land cover and urban areas in high resolution (1 km) is attached to this reply and will be added to the manuscript. All terrain, land cover and urban features are captured much better in the finer resolution, which is expected. However, it is not possible to run models on large domain as ours with such a fine resolution for a decade, because of high computational cost. Moreover, the major urban areas (like Berlin, Prague, Munich – analysed in our study) are equally well captured at 10 km resolution compared to the fine scale land cover. The location of chosen cities that are discussed in the study will be marked in the revised manuscript.

**Author's changes in manuscript:** Added new figure mapping the terrain elevation, land cover and urban areas as similar as Fig. 1 but rising from 1 km resolution static data, together with a caption and comment in the text. Figure 1 and its caption changed by marking of cities mentioned in the study.

**Referee's Comment #2:** Figure 9 and 10: Large uncertainties exist in urban-induced differences in vertical profiles of temperature and wind speed when compared across various configurations. But as also concluded by authors, lack of evaluation limits the ranking of the configurations used. In this regards, evaluation of the vertical profiles of temperature and wind speed against radiosonde observations (over or near these cities) could facilitate improvement in conclusions. Radiosonde observations over the domain of study are available openly from http://weather.uwyo.edu/upperair/sounding.html.

**Author's response:** The comparison of model vertical profiles of the temperature and wind speed with radiosonde observations are shown in attached figures. Because of the fact that only one measurement per city are available, only the simulations that match the real case (i.e. that include urban surfaces) are evaluated, not differences between urban and no-urban simulations, as in Fig. 9 and 10.

**Author's changes in manuscript:** Both attached figures will be added in the revised manuscript, together with comments in sections 2.3 (Observational data), 3.1 (Model

validation) and 4 (Discussion).

**Referee's Comment #3:**   Figure 11 and 12: Showing the percentage change in SI and VI due to urban surface will better underline the significant of the urban-induced differences in VI and SI.

**Author's response:**   As written in section 3.3, VI and SI distributions do not follow Gaussian distribution and it is not reasonable to represent the entire distribution by one value: the average is not correct for non-Gaussian distribution and modus or median are unrepresentative due to bi-modality in the SI distribution. The percentage change of the entire distribution would have very high range – from 0 to infinity (in terms of distribution shifts).

**Author's changes in manuscript:**   This is clarified in the revised manuscript.

**Referee's Comment #4:**   The conclusion that urban-induced modification enhances pollution dispersion is mainly based on the analysis over Prague. Authors should also check over other big cities in Europe to illustrate robustness of this association.

**Author's response:**   The author made the VI and SI distributions for discussed seasons and cities as Berlin, Munich and Budapest, but the characteristics of distributions and their changes are nearly the same, so they are not presented in the manuscript. The VI and SI distributions for Berlin are added, to show this fact.

**Author's changes in manuscript:**   Added sentence about VI and SI over cities: For other mentioned cities, the VI and SI distribution are nearly the same as in terms of Prague, thus only the Prague data are showed and discussed.

**Referee's Comment #5:**   Many short-term urban sensitivity simulations around the globe have shown that urban surfaces enhance convergence of low level horizontal wind over city center (Shepherd et al., 2005; Lin et al., 2008; Sarangi et al., 2018;

Niyogi et al., 2017; Zhong et al., 2017). This process can enhance the advection of particulate matter towards city center. Please include analysis/discussion about relative changes in convergence compared to VI and SI for these decade scale runs.

**Author's response:** All listed studies use high-resolution models where the vertical velocity and thus convergence is explicitly computed. In our study, at 10 km horizontal resolution with hydrostatic approximation, only the large-scale vertical velocity is computed, while sub-grid scale vertical motion is parametrized, so models need compensate only large-scale vertical motions by the horizontal convergence. For this reason, the convergence is not correctly captured in our study and thus is not possible to correctly evaluate it. On the other hand, the impact of cities on the vertical velocity (or convection) can be expressed by SI (Fig. 12), which tends to be higher in cities, thus also the convection is expected to be more intensive, with positive impact on the pollution via enhancing vertical transport from the boundary layer.

**Referee's Comment #6:** Also, the impact of urban surfaces on vertical velocity should be analysed/discussed in context to the urban-induced changes in VI simulated.

**Author's response:** As written above, we are not able to reasonably analyse the urban impact on the vertical velocity, because only the large-scale component of the vertical velocity is computed.

Shepherd, J. M. (2005). A review of current investigations of urban-induced rainfall and recommendations for the future. Earth Interactions,9(12), 1–27. https://doi.org/10.1175/EI156.1

Lin, C.-Y., F. Chen, J Huang, Y. A. Liou, W.C. Chen, W.N. Chen, and Shaw C. Liu, 2008: Urban heat island effect and its impact on boundary layer development and land-sea circulation over Northern Taiwan, Atmos. Environ., 42,5639-5649

Sarangi, C., Tripathi, S. N., Qian, Y.,Kumar, S., Ruby Leung, L. (2018).Aerosol and

urban land use effect onrainfall around cities in Indo-GangeticBasin from observations and cloudresolving model simulations. Journal ofGeophysical Research: Atmospheres, 123,3645–3667. https://doi.org/10.1002/2017JD028004

Niyogi, D., Lei, M., Kishtawal, C., Schmid, P., Shepherd, M. (2017). Urbanization impacts on the summer heavy rainfall climatology over theEastern United States. Earth Interactions, 21(5), 1–17. https://doi.org/10.1175/EI-D-15-0045.1

Zhong, S., Qian, Y., Zhao, C., Leung, R., Yang, X.-Q. (2015). A case study of urbanization impact on summer precipitation in the greaterBeijing metropolitan area: Urban heat island versus aerosol effects. Journal of Geophysical Research: Atmospheres, 120, 10,903-10,914.https://doi.org/10.1002/2015JD023753

——————————————————————

Elevation                                                                 m

Fig. 1. Improved Fig. 1

Elevation                                          m

Fig. 2. Added figure, as Fig. 1 but from 1 km resolution static data

[Figure]

Temperature vertical profiles above

Fig. 3. Added figure, temperature profile comparison with radiosonde observations

[Figure]

[Figure]

**Fig. 4.** Added figure, wind speed profile comparison with radiosonde observations

[Figure]

**Fig. 5.** The VI distribution in Berlin

[Figure]

**Fig. 6.** The SI distribution in Berlin